# Three Countries, Three Continents: AI Medical Device Regulation and Certification Comparison in the US, EU, and South Korea

**Abstract.** The rapid integration of artificial intelligence into medical devices has created unprecedented regulatory challenges, requiring novel certification frameworks that balance innovation with patient safety across major global markets. This paper provides a comparative analysis of AI medical device certification pathways in three leading jurisdictions: the United States (FDA), European Union (AI Act and MDR/IVDR), and South Korea (Medical Devices Act), examining risk classification systems, approval processes, and post-market requirements through systematic review of regulatory frameworks and guidance documents. Although all jurisdictions adopt risk-based approaches, implementation strategies vary significantly. The FDA emphasizes adaptive frameworks with predetermined change control plans, the EU combines horizontal AI Act requirements with sector-specific medical device regulations, and South Korea introduces novel high-impact AI provisions with incentivized impact assessments for medical applications. These regulatory differences create complex compliance landscapes for global manufacturers, making understanding of jurisdiction-specific pathways essential for strategic market entry and product development. The analysis in this paper provides actionable insights for optimizing certification strategies while ensuring patient safety across these critical markets.

**Keywords:** Artificial Intelligence · Medical Devices Regulation · EU AI Act · EU MDR · EU IVDR · Risk based Regulation.

## 1 Introduction and Need for regulation in AI based medical devices

Artificial intelligence is transforming modern medicine through enhanced diagnostic accuracy, accelerated drug discovery, and personalized treatment protocols. AI systems now assist radiologists in early cancer detection, predict patient deterioration, and automate administrative workflows across clinical settings. However, the sophisticated and adaptive nature of AI systems creates regulatory challenges that traditional medical device frameworks cannot adequately address. This has prompted major global health authorities to develop comprehensive AI-specific governance structures for medical devices, as documented by

Onitiu et al [26]. Real-world deployment failures demonstrate the inadequacy of conventional regulatory approaches for data-driven technologies. The IDx-DR system, among the first FDA-cleared autonomous AI systems for diabetic retinopathy screening, showed strong trial results but experienced significant performance degradation in practice—specificity dropped to 82% and positive predictive value fell to just 19%, causing excessive unnecessary referrals. This case highlighted the limitations of static premarket evaluation and underscored the need for adaptive regulatory frameworks [12]. Analysis of 266 safety events with FDA-approved ML devices revealed that 66% posed potential harm and 16% caused actual patient injury, with data input issues contributing to 82% of problems. While device malfunctions were most common, user-related problems were four times more likely to cause harm, emphasizing the need for comprehensive system-wide approaches to ML device safety [24]. Recent comparative analysis of AI medical device regulation demonstrates that each jurisdiction's approach reflects distinct technological maturity levels and cultural norms, precluding a one-size-fits-all regulatory standard [30]. This underscores the critical need for balanced regulatory frameworks that reconcile compliance requirements with objectives of ensuring safety, efficacy, and innovation in AI medical devices [30]. This comparative analysis focuses on three pivotal jurisdictions—the United States as the world's largest medical device market, the European Union with its pioneering AI Act, and South Korea as the first Asian nation to enact comprehensive AI legislation—representing distinct regulatory philosophies that collectively influence global AI governance standards. The following sections examine each jurisdiction's certification pathways, risk classification frameworks, and post-market surveillance requirements to identify best practices and regulatory gaps that will inform future AI medical device policy worldwide.

## 2   Definition of Medical Devices and AI System

The official definitions presented in Tables 1 and 2 demonstrate remarkable consistency in defining both AI systems and medical devices across jurisdictions, enabling meaningful comparison of how each jurisdiction translates these common definitions into distinct certification pathways.

## 3   AI Specific Regulations

The following comparison draws insights from established comparative studies [22] while focusing on key regulatory distinctions between the jurisdictions. While the EU established comprehensive AI regulation through its horizontal legislative approach with the AI Act entering force in August 2024, South Korea quickly followed as the first Asian jurisdiction to enact AI-specific legislation with its Basic Act passed in December 2024. Both regulatory frameworks emerged from similar policy motivations—balancing innovation promotion with risk mitigation—but reflect distinct regional approaches to AI governance, as detailed in Table 3. The EU's strict regulatory model emphasizes prohibited practices and

**Table 1.** Official Definitions of "AI System" in the US, EU, and South Korea

| Region | Official Definition of "AI System" |
|---|---|
| European Union | "AI system" means a machine-based system that is designed to operate with varying levels of autonomy and that may exhibit adaptiveness after deployment, and that, for explicit or implicit objectives, infers, from the input it receives, how to generate outputs such as predictions, content, recommendations, or decisions that can influence physical or virtual environments. [6] |
| United States | A machine-based system that can, for a given set of human-defined objectives, make predictions, recommendations, or decisions influencing real or virtual environments. Artificial intelligence systems use machine- and human-based inputs to perceive real and virtual environments; abstract such perceptions into models through analysis in an automated manner; and use model inference to formulate options for information or action. [33] |
| South Korea | An artificial intelligence-based system that infers results such as predictions, recommendations and decisions that affect real and virtual environments for a given goal with various levels of autonomy and adaptability. [27] |

**Table 2.** Official Legal Definitions of Medical Device in EU, US and South Korea

| Region | Official Definition of Medical Device |
|---|---|
| European Union | "medical device" means any instrument, apparatus, appliance, software, implant, reagent, material or other article intended by the manufacturer to be used, alone or in combination, for human beings for medical purposes: diagnosis, prevention, monitoring, prediction, prognosis, treatment or alleviation of disease; diagnosis, monitoring, treatment, alleviation of, or compensation for, an injury or disability; investigation, replacement or modification of the anatomy or physiological/pathological process; providing information by means of in vitro examination of specimens derived from the human body; and which does not achieve its principal intended action by pharmacological, immunological or metabolic means. Products for conception control and cleaning/disinfection/sterilisation of medical devices are also included. [1] |
| United States | The term "device" means an instrument, apparatus, implement, machine, contrivance, implant, in vitro reagent, or similar article, including component parts or accessories which is: (A) recognized in official formularies, (B) intended for diagnosis, cure, mitigation, treatment, or prevention of disease, or (C) intended to affect body structure or function, and which does not achieve primary purposes through chemical action or metabolism. [21] |
| South Korea | The term "medical device" means an instrument, machine, apparatus, material, software, or similar product used alone or in combination for: 1. diagnosing, curing, alleviating, treating, or preventing disease; 2. diagnosing, curing, alleviating, or correcting injury or impairment; 3. testing, replacing, or transforming structure or function; 4. control of conception. [3] |

substantial penalties, while South Korea adopts a more industry-collaborative approach focused on supporting domestic AI development alongside measured oversight. These parallel developments in 2024 demonstrate the global momentum toward establishing comprehensive AI regulatory frameworks, with both

acts serving as influential models for other jurisdictions considering AI legislation.

**Table 3.** Comparison of EU AI Act and South Korea AI Basic Act

| Aspect | EU AI Act | South Korea AI Basic Act |
|---|---|---|
| **Legal Framework** | Regulation (EU) 2024/1689 - Entered into force August 1, 2024, with staggered implementation: prohibited practices (February 2, 2025), GPAI obligations (August 2, 2025), full application (August 2, 2026) [5] | Basic Act on the Development of Artificial Intelligence and Establishment of Trust - Passed December 26, 2024, taking effect January 22, 2026 [29] |
| **Risk Classification** | Four-tier system: (1) Prohibited AI practices (Article 5), (2) High-risk AI systems (Article 6), (3) Limited risk AI with transparency obligations, (4) General-purpose AI models with systemic risk [5] | Risk-based approach with two main categories: (1) High-impact AI systems (Article 2), (2) Generative AI systems, plus obligations for AI systems exceeding computational thresholds [29] |
| **Prohibited Practices** | Article 5 explicitly bans eight AI practices including: cognitive behavioral manipulation, social scoring, real-time biometric identification in public spaces (with limited exceptions), and AI systems predicting future criminality [5] | No explicit prohibition of AI practices. Instead focuses on ensuring safety and reliability of high-impact AI and transparency regulations for generative AI [29,27] |
| **Penalties** | Maximum penalties: €35 million or 7% of global annual turnover for prohibited AI practices (Article 99(3)); €15 million or 3% for other violations; €7.5 million or 1% for providing misleading information [5] | Fines up to KRW 30 million (~$20,870 USD) for violations. Significantly lower than EU penalties [29] |
| **Governance** | European AI Office within European Commission for oversight (Article 65), AI Board for coordination (Article 66), Scientific Panel of independent experts (Article 68), and national competent authorities (Article 70) [5] | National AI Committee chaired by the President (Article 7), AI Policy Center (Article 11), AI Safety Research Institute, and Korea AI Promotion Association for comprehensive governance framework [29] |

# 4  Overview of Regulatory Frameworks for AI Medical Devices

This section examines the regulatory approaches of three leading jurisdictions for AI-powered medical devices. Table 4 provides a detailed comparison of their frameworks.Additionally, data protection requirements vary across jurisdictions: the EU enforces GDPR compliance for medical AI systems [13], South Korea applies the Personal Information Protection Act (PIPA) [28], while the US lacks comprehensive federal data privacy legislation specific to medical devices beyond HIPAA for covered entities [32].

*European Union* The EU operates dual regulatory compliance: MDR/IVDR for medical device approval and the 2024 AI Act for AI system oversight [**?**]. The AI Act classifies systems into four risk levels (unacceptable, high, limited, minimal risk), with medical devices Class IIa and above automatically designated as high-risk AI systems requiring additional conformity assessments [9].

*United States* The FDA utilizes risk-based pathways (510(k), De Novo, PMA) with Predetermined Change Control Plans (PCCPs) enabling pre-authorized modifications for adaptive AI systems without requiring new submissions [34].

*South Korea* South Korea's MFDS led comprehensive digital health regulation through the Digital Medical Products Act (2024), establishing the world's first framework specifically for digital medical products including AI devices [25].

**Table 4.** High-level comparison of regulatory frameworks for AI in medical devices.

| Attribute | European Union | United States | South Korea |
|---|---|---|---|
| **Lead Agency** | European Commission, NCAs, NBs | FDA | MFDS |
| **Core Medical Device Law** | MDR (EU 2017/745); IVDR (EU 2017/746) | FD&C Act; 21 CFR 807, 814, 860 | Medical Device Act; DMPA (2024) |
| **AI-Specific Regulation** | AI Act (EU 2024/1689) high-risk systems; MDR/IVDR compliance | FDA Final Guidance for AI-DSF (Dec 2024); PCCP framework | DMPA (2024); AI/ML guidelines; Generative AI guideline (2025) |
| **Risk Classification** | Class I–III (MDR); A–D (IVDR); AI Act high-risk designation | Class I–III with AI change management | Class 1–4 with algorithm maturity criteria |
| **Approval Pathways** | CE Marking via NB; Dual MDR/AI Act conformity | 510(k), De Novo, PMA + PCCP | Standard/expedited (80-day); Third-Party Reviewer |
| **Post-Market Requirements** | Vigilance (MDR/IVDR), Art. 73 (AI Act) reporting | MDR, QSR, real-world monitoring, PCCP updates | Adverse event reporting, DMPA monitoring, cybersecurity |
| **AI/Software Definitions** | MDSW (MDR/IVDR); AI system (Art. 3) | IMDRF SaMD/SiMD; AI-DSF | AIMD; Digital medical products (AI/ML) |

## 5   Risk Class in Medical Devices

Risk classification systems fundamentally shape the commercial viability and development trajectory of AI-enabled medical devices, where higher risk classifications require exponentially more resources and longer development timelines compared to lower-risk classifications. As shown in Table 5, the United States operates a three-tier system (Classes I, II, III) while South Korea employs a four-tier system (Classes I, II, III, IV), creating strategic decisions about market entry sequencing for global manufacturers. Companies often pursue regulatory arbitrage by initially developing products to meet requirements of markets with less restrictive pathways, then modifying products to satisfy more stringent regulatory frameworks. The European Union presents unique regulatory complexity

through its dual classification system [8], where AI-enabled medical devices must simultaneously satisfy traditional medical device regulations (MDR/IVDR) and AI Act requirements, a layered compliance burden not present in other jurisdictions. South Korea's four-tier classification system includes a Class IV category for the highest-risk devices, while the US and EU address similar high-risk devices within their existing three-tier frameworks, creating different regulatory pathways for equivalent risk levels. The absence of harmonization across these classification systems means companies cannot transfer approval evidence between jurisdictions, necessitating region-specific development and validation strategies that significantly impact development timelines and regulatory costs.

**Table 5.** Risk Classification of AI-Based Medical Devices in South Korea, United States, and European Union

| Region | Risk Classes & Criteria |
| --- | --- |
| **European Union [1,2,5]** | **MDR:** Class I (Low Risk), IIa/IIb (Moderate/High Risk), III (Highest Risk).
**IVDR:** Class A (Low Risk), B, C, D (Highest Risk).
**AI Act High-Risk:** Any AI system that is a medical device or IVD requiring Notified Body review under MDR/IVDR is automatically "high-risk" under AI Act Article 6 and Annex II. |
| **United States[21]** | **Class I (Low Risk):** General controls.
**Class II (Moderate Risk):** 510(k) clearance.
**Class III (High Risk):** Pre Market Approval required. |
| **South Korea [14]** | **Class 1 (Low Risk):** Non-invasive, minimal risk.
**Class 2 (Moderate Risk):** Moderate risk, non-critical diagnostics.
**Class 3 (High Risk):** Critical diagnostics.
**Class 4 (Highest Risk):** Life-sustaining/autonomous AI. |

# 6   Regulatory Approval of AI-Enabled Medical Devices: Requirements and Pathways by Region

Table 6 compares certification requirements for AI-based medical devices across the United States (FDA), European Union (MDR/IVDR with AI Act), and South Korea (MFDS with Digital Medical Products Act). The US uses a 3-tier classification system (Class I-III) while the EU and South Korea utilize 4-tier frameworks. Regulatory approaches differ significantly: the US relies on FDA pathways with Predetermined Change Control Plans (PCCP) for adaptive AI systems; the EU requires dual compliance with medical device regulations and the AI Act, including mandatory CE marking and automatic high-risk

classification for devices requiring Notified Body assessment; South Korea implemented the Digital Medical Products Act (effective January 24, 2025) with specific cloud infrastructure and cybersecurity requirements. All jurisdictions require algorithm description and post-market surveillance. Table 6 provides specific legal references and article numbers to enable manufacturers to navigate regulatory pathways for AI medical device market access.

## 7   Insights and Conclusion

Regulatory affairs professionals must navigate unprecedented complexity with EU dual compliance (MDR + AI Act) while US PCCP frameworks enable pre-authorized algorithm updates. Academic researchers and policy makers should prioritize regulatory science development for adaptive AI validation methodologies and international harmonization frameworks addressing fundamental technology governance questions. Healthcare practitioners require enhanced AI literacy training to understand system limitations, maintain clinical oversight authority, and implement appropriate patient communication strategies for AI-assisted medical decisions.

Procedurally, the US leverages 510(k) substantial equivalence (96.5% of AI devices) with 133-day median clearance, EU requires CE marking through notified body assessment combining medical device and AI Act requirements, while Korea's DMPA provides streamlined 30-80 day pathways with substantial equivalence options. Philosophically, the US operates on market-oriented innovation prioritizing speed and substantial equivalence, the EU implements precautionary dual-framework approaches emphasizing data protection and fundamental rights, while South Korea adopts process-oriented systematic evaluation balanced with innovation support. Beyond these three regions, China has approved 59 AI medical devices with comprehensive lifecycle guidelines integrated with national AI development strategy [7], while Japan's PMDA offers 6-month priority reviews through Sakigake fast-track systems [4].

The EU Data Act becomes effective September 12, 2025 [31], requiring IoT medical device data sharing with potential conflicts between transparency obligations and proprietary AI algorithm protection, while FDA's January 6, 2025 [20] comprehensive draft guidance on AI-enabled device lifecycle management represents the first complete regulatory framework for adaptive medical AI systems.

## 8   Funding Information

If accepted -Project Funding and grant details added here. —

**Table 6.** Certification Pathways for AI-Based Medical Devices: US, EU, South Korea

| Region | Certification Pathway | Legal Basis / Article Numbers |
|---|---|---|
| **European Union (MDR/IVDR + CE Marking + AI Act)** | **MDR:**
Class I: Self-declaration by manufacturer.
Class IIa/IIb/III: Notified Body review required.
**IVDR:**
Class A: Self-declaration.
Class B/C/D: Notified Body review required.
**CE Marking:** All medical devices and IVDs must bear the CE mark after successful conformity assessment (MDR Art. 20, IVDR Art. 18, AI Act Art. 48).
**AI Act:** Any device requiring Notified Body assessment under MDR/IVDR is automatically "high-risk" (AI Act Art. 6, Annex II) and must comply with all AI Act obligations (risk management, transparency, human oversight, etc.).
**Post-market:** Surveillance required under MDR/IVDR (Art. 83–86 MDR; Art. 78–81 IVDR) and AI Act (Art. 72–73). | MDR: Art. 52, Annex VIII[1]
IVDR: Art. 48, Annex VIII[2]
CE Marking: MDR Art. 20, IVDR Art. 18, AI Act Art. 48[10,5]
AI Act: Art. 6, Annex II, Art. 48[5] |
| **United States (FDA)** | **Class I:** General controls. Most are exempt from premarket notification (510(k)).
**Class II:** 510(k) premarket notification showing substantial equivalence to a predicate device. Most AI/ML-enabled SaMD are Class II.
**Class III:** Premarket Approval (PMA) required for high-risk devices; extensive clinical evidence required.
**AI-specific:** For adaptive AI, a Predetermined Change Control Plan (PCCP) must be submitted. All AI-enabled devices must include algorithm description, validation, and performance metrics.
**Post-market:** Surveillance required for all classes; PCCPs allow some pre-authorized software updates. | FD&C Act, 21 U.S.C. §360c (Section 513)[11]
21 CFR Part 860[19], 807[17], 814[18]
FDA AI/ML Guidance (2024)[20] |
| **South Korea (MFDS, DMPA)** | **Class I:** Notification pathway; no clinical data required.
**Class II:** Pre-market certification; performance data required.
**Class III:** Pre-market approval; clinical validation required.
**Class IV:** Pre-market approval; clinical validation and Summary Technical documentation required.
**AI-specific:** Algorithm description, training/validation data, and cybersecurity documentation required.
**Post-market:** Ongoing monitoring and reporting required for all classes, especially for adaptive AI and Class 4 devices.
**Latest Acts:** The Digital Medical Products Act (DMPA, effective January 24, 2025) primarily regulates digital medical products, with the Medical Devices Act applying where DMPA has no corresponding provisions. | MFDS Medical Device Act, "Classification of medical devices"[16]
MFDS Approval Process Overview[15]
MFDS AI-Based Medical Device Guidance[14]
DMPA: Digital Medical Products Act (2025)[23] |

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
