# OpenReview forum: "Three Countries, Three Continents: AI Medical Device Regulation and Certification Comparison in the US, EU, and South Korea"
_MICCAI.org/2025/Workshop/BRIDGE — BRIDGE 2025 Poster_

### Official Review · Reviewer_uEDm · 2025-07-23
**Comprehensive Regulatory Mapping Lacks Strategic Solutions for AI Medical Device Implementation**

**Rating:** 7
**Confidence:** 4

**Review:**

This paper provides a nice comparison of current state of AI medical device regulation across three major jurisdictions. Its comprehensive mapping of regulatory frameworks and attention to recent developments make it a useful resource for manufacturers, researchers, and policymakers seeking to understand the complex global regulatory landscape.
However, its impact is limited by a predominantly descriptive approach lacking deeper analytical insights. We recommend adding at least one page of solution-focused analysis, as the workshop aims to address practical regulatory and deployment challenges rather than merely document existing frameworks.

 This paper is relevant to the BRIDGE workshop and can ignite interesting discussion.

---

### Official Review · Reviewer_wW7v · 2025-07-25
**Comparative studies of regulatory pathways across regions without any key insights**

**Rating:** 5
**Confidence:** 4

**Review:**

## Summary of the Paper
This paper provides a comparative analysis of AI medical device certification pathways across three major jurisdictions: the United States (FDA), European Union (AI Act and MDR/IVDR), and South Korea (Medical Devices Act and Digital Medical Products Act). The authors examine risk classification systems, approval processes, and post-market requirements, serving as a reference guide that consolidates publicly available regulatory information across these three jurisdictions into a single comparative framework.

## Strengths
- Comprehensive regulatory framework mapping: The paper provides detailed side-by-side comparisons of three major jurisdictions with specific legal references, article numbers, and implementation timelines that would be valuable for manufacturers navigating global market entry.
- Timely analysis of emerging regulations: The authors capture very recent regulatory developments including the EU AI Act (August 2024), South Korea's AI Basic Act (December 2024), and FDA's updated AI guidance (January 2025), providing current information for industry decision-making.
- Practical focus on manufacturer challenges: The paper explicitly addresses real-world compliance burdens such as dual EU compliance requirements (MDR + AI Act) and regulatory arbitrage opportunities, demonstrating understanding of industry pain points.

## Limitations or Areas for Improvement
- Lack of proper citations for key claims: Critical statements lack supporting references, such as "Companies often pursue regulatory arbitrage by initially developing products to meet requirements of markets with less restrictive pathways" (Section 5) - this claim requires citation to industry reports or regulatory guidance documents.
- Inaccurate statements about evidence transferability: The claim that "The absence of harmonization across these classification systems means companies cannot transfer approval evidence between jurisdictions" (Section 5) is incorrect. Clinical trial evidence can sometimes be used across jurisdictions with appropriate justification regarding intended use population and risk-based assessments.
- Limited analytical insights beyond fact compilation: While the paper organizes publicly available information effectively, it lacks groundbreaking insights or identification of common themes connecting the three jurisdictions. The analysis remains largely descriptive rather than providing strategic guidance for manufacturers.
- Missing synthesis of actionable patterns: The paper presents a bunch of facts organized across sections without synthesizing these into meaningful strategic recommendations or identifying underlying regulatory principles that could guide manufacturer decision-making.

## Relevance to BRIDGE Workshop Topics
- Comparative studies of regulatory pathways across regions: provides systematic comparison framework
- Early regulatory engagement strategies: offers baseline knowledge for planning. However, it falls short of the workshop's emphasis on novel insights, empirical rigor, or innovative approaches to regulatory challenges.

---

### Official Review · Reviewer_cWNR · 2025-07-26
**Review Comments**

**Rating:** 6
**Confidence:** 4

**Review:**

This paper provides a comparative analysis of AI medical device regulations across three major jurisdictions (US, EU, South Korea). The systematic comparison is well-structured and informative.

Paper Strengths:
The paper thoroughly examines regulatory frameworks across three strategically important jurisdictions.
Tables and systematic comparisons make the information accessible and easy to understand.

However, the paper has several concerns that need to be addressed.
Several key claims lack proper citations or have unclear referencing. For example, "Analysis of 266 safety events with FDA-approved ML devices revealed that 66% posed potential harm and 16% caused actual patient injury, with data input issues contributing to 82% of problems." Multiple assertions about jurisdiction selection rationale and global influence lack adequate supporting references. Please provide appropriate references and ensure all factual claims are properly attributed and cited. Also, please fix the issues with the reference list, as it is exceeding the page limit.
The paper reads more like a regulatory handbook than an academic research contribution. Specifically, the analysis focuses almost exclusively on describing existing regulations rather than critically evaluating regulatory effectiveness, discussing challenges, or proposing solutions. Please enhance the paper by transforming descriptive content to focus more on challenges and solutions.
The paper should include a substantial section analyzing specific limitations within each regulatory system. For example, what are FDA-Specific Limitations, EU-Specific Limitations, and South Korea-Specific Limitations? And what are Cross-Jurisdictional Limitations (e.g., no standardized approach to regulating emerging AI technologies such as foundation models in healthcare)?
The paper lacks Forward-Looking Solutions discussion. Please include a dedicated "Recommendations and Solutions" section addressing challenges such as Regulatory Harmonization, Technological Challenges, etc.
Please expand the paper by discussing the issues listed above.

---

### Decision · Program_Chairs · 2025-07-26

**Decision:**

Accept (Poster)

**Comment:**

Dear Authors,

We inform you that your paper is provisionally accepted, provided you comprehensively address the reviewer comments below. To secure final acceptance, please revise accordingly and submit your updated manuscript by July 31.

Requirements for your final submission:

- Address all reviewer comments throughout your paper; at minimum, add a discussion section that explicitly acknowledges and responds to each key point raised.
- Ensure your final draft adheres to Springer formatting guidelines.
- Submit the source file along with any supplementary material.

Best regards,
BRIDGE Workshop Organizers